# Comparison and Assessment of Regional and Global Land Cover Datasets for Use in CLASS over Canada

**Libo Wang** [1],*, **Paul Bartlett** [1], **Darren Pouliot** [2], **Ed Chan** [1], **Céline Lamarche** [3], **Michael A. Wulder** [4], **Pierre Defourny** [3] and **Mike Brady** [1]

[1] Climate Research Division, Environment and Climate Change Canada, 4905 Dufferin Street, Toronto, ON M3H 5T4, Canada; Paul.Bartlett@canada.ca (P.B.); Ed.chan@canada.ca (E.C.); Mike.Brady@canada.ca (M.B.)

[2] Landscape Science and Technology, Environment and Climate Change Canada, 1125 Colonel By Drive, Ottawa, ON K1S 5B6, Canada; Darren.Pouliot@canada.ca

[3] Earth and Life Institute, Environmental Sciences, Université catholique de Louvain, Croix du Sud, 2-L7.05.16, 1348 Louvain-la-Neuve, Belgium; celine.lamarche@uclouvain.be (C.L.); Pierre.Defourny@uclouvain.be (P.D.)

[4] Canadian Forest Service (Pacific Forestry Centre), Natural Resources Canada, 506 West Burnside Road, Victoria, BC V8Z 1M5, Canada; Mike.Wulder@canada.ca

*  Correspondence: Libo.Wang@canada.ca; Tel.: +1-416-664-5358

**Abstract:** Global land cover information is required to initialize land surface and Earth system models. In recent years, new land cover (LC) datasets at finer spatial resolutions have become available while those currently implemented in most models are outdated. This study assesses the applicability of the Climate Change Initiative (CCI) LC product for use in the Canadian Land Surface Scheme (CLASS) through comparison with finer resolution datasets over Canada, assisted with reference sample data and a vegetation continuous field tree cover fraction dataset. The results show that in comparison with the finer resolution maps over Canada, the 300 m CCI product provides much improved LC distribution over that from the 1 km GLC2000 dataset currently used to provide initial surface conditions in CLASS. However, the CCI dataset appears to overestimate needleleaf forest cover especially in the taiga-tundra transition zone of northwestern Canada. This may have partly resulted from limited availability of clear sky MEdium Resolution Imaging Spectrometer (MERIS) images used to generate the CCI classification maps due to the long snow cover season in Canada. In addition, changes based on the CCI time series are not always consistent with those from the MODIS or a Landsat-based forest cover change dataset, especially prior to 2003 when only coarse spatial resolution satellite data were available for change detection in the CCI product. It will be helpful for application in global simulations to determine whether these results also apply to other regions with similar landscapes, such as Eurasia. Nevertheless, the detailed LC classes and finer spatial resolution in the CCI dataset provide an improved reference map for use in land surface models in Canada. The results also suggest that uncertainties in the current cross-walking tables are a major source of the often large differences in the plant functional types (PFT) maps, and should be an area of focus in future work.

**Keywords:** land cover; forest cover; plant functional type; land surface model; CLASS; Canada

## 1. Introduction

Accurate characterization of land cover and related dynamics are required to document environmental changes and to provide initial surface conditions for land surface, ecosystem, climate, and Earth system models [1,2]. Land cover (LC) is considered an essential climate variable due to its influence on the exchange of energy, water, and carbon between the land surface and the

atmosphere [3,4]. Satellite remote sensing provides a critical data source for LC mapping from regional to global scales [5,6]. Earlier global LC products, such as the International Geosphere-Biosphere Programme Data and Information System land Cover data set [7], the University of Maryland Land Cover Classification [8], and the Global Land Cover 2000 (GLC2000) data set [9], were based on optical bands either from the Advanced Very High Resolution Radiometer (AVHRR) series or SPOT-VEGETATION (VEG) at 1 km resolution. The development of satellite optical sensors with more spectral bands and finer spatial resolutions enabled the creation of more detailed global LC products, such as the Moderate Resolution Imaging Spectroradiometer (MODIS) LC dataset at 500 m resolution [10], as well as the GlobCover and the Climate Change Initiative (CCI) LC datasets at 300 m resolution [11,12]. Cihlar [13] documented this history of LC mapping and provided insights regarding the transition and rationale for map production with higher spatial resolution data sources. Townshend [14] related the state-of-the-art at that time, with the desire for an increase in spatial resolution noted by [13]. The increase in spatial resolution was foreseen to improve map quality and detail, but to also reduce mixed pixels and aid with cloud screening. As a follow-on, Wulder et al. [15] describe the current nature of LC mapping, with free and open datasets provided in increasingly analysis-ready forms, improved computing opportunities and algorithmic options, as well as reduced production times.

To ensure confidence in any maps produced, significant efforts have been made to collect reference data to develop and assess regional and global LC products, which often directly or indirectly involves the interpretation of higher resolution satellite or airborne images [16–18]. Based on existing reference sample sites across the globe, Herold et al. [19] showed that the earlier 1 km resolution LC datasets had limited ability to discriminate mosaic vegetation classes characterized by assemblages of forests, shrubs, and grasslands. In addition, there was a close correlation between classification accuracy and landscape homogeneity, which was defined as the fraction occupied by the dominant LC type in a unit area. The maximum achievable accuracy was dependent on the spatial and thematic resolution and heterogeneity of landscapes [16]. The finer 300 m spatial resolution of the recent GlobCover and CCI LC products makes them inherently superior for LC mapping in heterogeneous landscapes where different datasets tend to disagree. As such, it would be preferable to replace older LC products with finer spatial resolution to initialize surface conditions in the various models. Tsenbazar et al. [20] argue that the accuracy of LC products needs to be assessed for the specific application or intended use.

Most land surface models (LSMs) represent global vegetation using a set of plant functional types (PFTs), the number and type of which differ in each LSM [1]. For example, the Canadian Land Surface Scheme (CLASS) recognizes four broad PFTs, needleleaf forests (NF), broadleaf forests (BF), crops and grass. CLASS is a physically-based LSM [21,22], incorporating complete thermal and hydrological budgets, including canopy snow processes such as interception, unloading, sublimation and melt [23,24]. The carbon cycle, related ecosystem processes and dynamic vegetation can be simulated by coupling CLASS with the Canadian Terrestrial Ecosystem Model (CTEM) [25–27], which increases the number of PFTs recognized to nine, as subsets of the initial four in CLASS. Table 1 in [26] showed how the nine PFTs in CTEM can be mapped onto the four in CLASS.

Because different PFTs can be involved with different model processes and parameters, the spatial distribution and fractional cover of PFTs are important for accurate simulation of carbon, water and energy budgets [4,28,29]. For example, the surface albedos for needleleaf evergreen trees, broadleaf deciduous trees, crops, and grasslands can be very different, especially during winter when deciduous trees are leafless and short vegetation is largely buried by snow [24,30]. Surface roughness for short or tall vegetation is also very different, and this affects simulated turbulent exchanges. Uncertainties in PFT mapping can be linked to the uncertainties in LC datasets and/or the cross-walking (CW) tables used to convert the LC classes to PFT fractions [31]. The distribution of PFTs in CLASS when employed as the land surface component of the Environment and Climate Change Canada (ECCC) climate and Earth system models is based on GLC2000 [32,33]. Wang et al. [34] found that the bias in winter albedo in selected boreal forest regions among the Coupled Model Intercomparison Phase 5 (CMIP5) models

was largely related to bias in leaf area index and tree cover fraction. They also showed that the March surface albedo in the ECCC atmospheric general circulation model (CanAM 4.2) was underestimated over much of the boreal forest. These results provided the impetus to assess the current GLC2000-based LC dataset with respect to newer alternatives.

**Table 1.** Legends for land cover datasets included in this study. The parentheses in the CCI legend are used to indicate sub-classes, while the brackets show the main classes.

| NALCMS/MODIS | EOSD | CCI | GLC2000 |
|---|---|---|---|
| [1] Temperate or sub-polar needleleaf forest | [11] Cloud | [10] Cropland rainfed | [1] Tree cover, broadleaved, evergreen |
| [2] Sub-polar taiga needleleaf forest | [12] Shadow | (11) Herbaceous cover | [2] Tree cover, broadleaved, deciduous, closed |
| [3] Tropical or sub-tropical broadleaf evergreen forest | [20] Water | (12) Tree or shrub cover | [3] Tree cover, broadleaved, deciduous, open |
| [4] Tropical or sub-tropical broadleaf deciduous forest | [31] Snow/Ice | [20] Cropland irrigated or post-flooding | [4] Tree cover, needle-leaved, evergreen |
| [5] Temperate or sub-polar broadleaf deciduous forest | [32] Rock/Rubble | [30] Mosaic cropland (> 50%) / natural vegetation (tree shrub herbaceous cover) (< 50%) | [5] Tree cover, needle-leaved, deciduous |
| [6] Mixed forest | [33] Exposed/Barren Land | [40] Mosaic natural vegetation (tree shrub herbaceous cover) (> 50%) / cropland (< 50%) | [6] Tree cover, mixed leaf type |
| [7] Tropical or sub-tropical shrubland | [40] Bryoids | [50] Tree cover broadleaved evergreen closed to open (> 15%) | [7] Tree cover, regularly flooded, fresh water |
| [8] Temperate or sub-polar shrubland | [51] Shrub Tall | [60] Tree cover broadleaved deciduous closed to open (> 15%) | [8] Tree cover, regularly flooded, saline water |
| [9] Tropical or sub-tropical grassland | [52] Shrub Low | (61) Tree cover broadleaved deciduous closed (> 40%) | [9] Mosaic: tree cover / other natural vegetation |
| [10] Temperate or sub-polar grassland | [81] Wetland-treed | (62) Tree cover broadleaved deciduous open (15–40%) | [10] Tree cover, burnt |
| [11] Sub-polar or polar shrubland-lichen-moss | [82] Wetland-shrub | [70] Tree cover needleleaved evergreen closed to open (> 15%) | [11] Shrub cover, closed-open, evergreen |
| [12] Sub-polar or polar grassland-lichen-moss | [83] Wetland-herb | (71) Tree cover needleleaved evergreen closed (> 40%) | [12] Shrub cover, closed-open, deciduous |
| [13] Sub-polar or polar barren-lichen-moss | [100] Herbs | (72) Tree cover needleleaved evergreen open (15–40%) | [13] Herbaceous cover, closed-open |
| [14] Wetland | [110] Grassland | [80] Tree cover needleleaved deciduous closed to open (>15%) | [14] Sparse herbaceous or sparse shrub cover |
| [15] Cropland | [211] Coniferous-dense | (81) Tree cover needleleaved deciduous closed (> 40%) | [15] Regularly flooded shrub and/or herbaceous cover |
| [16] Barren lands | [212] Coniferous-open | (82) Tree cover needleleaved deciduous open (15–40%) | [16] Cultivated and managed areas |
| [17] Urban | [213] Coniferous-sparse | [90] Tree cover mixed leaf type (broadleaved and needleleaved) | [17] Mosaic: cropland / tree cover / other natural veg. |
| [18] Water | [221] Broadleaf-dense | [100] Mosaic tree and shrub (> 50%) / herbaceous cover (< 50%) | [18] Mosaic: cropland / shrub and/or grass cover |
| [19] Snow and Ice | [222] Broadleaf-open | [110] Mosaic herbaceous cover (> 50%) / tree and shrub (< 50%) | [19] Bare areas |
| | [223] Broadleaf-sparse | [120] Shrubland | [20] Water bodies |
| | [231] Mixedwood-dense | (121) Shrubland evergreen | [21] Snow and ice |
| | [232] Mixedwood-open | (122) Shrubland deciduous | [22] Artificial surfaces and associated areas |
| | [233] Mixedwood-sparse | [130] Grassland | |
| | | [140] Lichens and mosses | |
| | | [150] Sparse vegetation (tree shrub herbaceous cover) (< 15%) | |
| | | (151) Sparse tree (< 15%) | |
| | | (152) Sparse shrub (< 15%) | |
| | | (153) Sparse herbaceous cover (< 15%) | |
| | | [160] Tree cover flooded fresh or brakish water | |
| | | [170] Tree cover flooded saline water | |
| | | [180] Shrub or herbaceous cover flooded fresh/saline/brakish water | |
| | | [190] Urban areas | |
| | | [200] Bare areas | |
| | | (201) Consolidated bare areas | |
| | | (202) Unconsolidated bare areas | |
| | | [210] Water bodies | |
| | | [220] Permanent snow and ice | |

The CCI LC datasets recently generated by the European Space Agency are available annually from 1992 to 2015 at 300 m resolution [12]. This dataset was produced based on broad user consultation, specifically to address the needs of the climate modelling community [35]. The CCI dataset is

considered a good candidate to provide more accurate PFTs than the current GLC2000 dataset used in CLASS [36]. The objective of this study is to assess and compare the CCI and the GLC2000 datasets with high-resolution (~30 m) land cover products over Canada, to better understand their patterns of agreement and disagreement, and to determine the applicability of the CCI dataset for use in CLASS. The evaluation focuses on forested areas, where the aforementioned inaccurate representation of vegetation distribution and parameters were linked to a large spread in simulated surface albedo and snow-albedo feedback strength in CMIP5 models [34,37,38].

## 2. Study Area and Materials

### 2.1. Study Area

The study area focuses on Canada, where several national LC products have been generated from satellite data sources. Canada is almost a billion hectares in area and forest dominated ecosystems occupy approximately 65% of the national land base. North of forested ecosystems are sparsely populated tundra and arctic environments. Southern Canada is dominated by a mosaic of agricultural and urban land use (Figure 1). We divide Canada into 18 regions determined by ecozone (EZ) [39] or sub-ecozone (Figure 1). Some of the large ecozones are sub-divided into smaller ones mainly based on the location and the spatial distribution of LC characteristics. For example, the Boreal Shield zone extends from northeastern Alberta to Newfoundland and Labrador, and is divided into five sub-zones (EZ5-9), with EZ5 dominated by mixed forest (MF), EZ6, EZ7 and EZ9 dominated by NF, and EZ8 by shrub. Names of the EZs can be found in Table 2.

**Table 2.** Percentage of wetland (Wland) and Wland-treed (based on ratio from EOSD) from NALCMS, ratio of taiga needleleaf vs. total needleleaf in NALCMS, and homogeneity in CCI (brackets show cover type) in 18 ecozones across Canada.

| ECOZONE | NALCMS Wland (%) | NALCMS Wland-Treed (%) | EOSD Ratio of Wland-Treed | NALCMS Ratio of Taiga NF/Total NF | CCI Homogeneity |
|---|---|---|---|---|---|
| EZ01_Atlantic_Highlands | 0.53 | 0.11 | 0.20 | 0.00 | 0.48 (MF) |
| EZ02_Atlantic_Maritime | 2.25 | 0.45 | 0.20 | 0.00 | 0.40 (NF) |
| EZ03_Boreal_Cordillera | 0.10 | 0.01 | 0.10 | 0.02 | 0.59 (NF) |
| EZ04_Boreal_Plains | 4.79 | 1.79 | 0.37 | 0.01 | 0.42 (NF) |
| EZ05_Boreal_Shield_South | 3.75 | 1.85 | 0.49 | 0.00 | 0.38 (mix) |
| EZ06_Boreal_Shield_Labrador | 2.27 | 0.38 | 0.17 | 0.01 | 0.74 (NF) |
| EZ07_Boreal_Shield_West | 11.06 | 4.47 | 0.40 | 0.06 | 0.68 (NF) |
| EZ08_Boreal_Shield_Newfoundland | 5.11 | 0.12 | 0.02 | 0.00 | 0.30 (shrub) |
| EZ09_Boreal_Shield_East | 0.90 | 0.07 | 0.08 | 0.01 | 0.64 (NF) |
| EZ10_Hudson_Plains | 37.17 | 18.31 | 0.49 | 0.43 | 0.38 (NF) |
| EZ11_Montane_Cordillera | 0.16 | 0.01 | 0.05 | 0.01 | 0.68 (NF) |
| EZ12_Prairies | 0.90 | 0.00 | 0.00 | 0.00 | 0.73 (crop) |
| EZ13_Southern_Arctic | 0.91 | 0.15 | 0.17 | 0.48 | 0.72 (sparse) |
| EZ14_Taiga_Cordillera | 0.53 | 0.05 | 0.10 | 0.07 | 0.30 (NF) |
| EZ15_Taiga_Plains_South | 6.14 | 2.11 | 0.34 | 0.21 | 0.56 (NF) |
| EZ16_Taiga_Plains_North | 2.03 | 0.21 | 0.10 | 0.49 | 0.55 (NF) |
| EZ17_Taiga_Shield_East | 0.86 | 0.14 | 0.16 | 0.15 | 0.53 (NF) |
| EZ18_Taiga_Shield_West | 4.92 | 1.26 | 0.26 | 0.31 | 0.42 (NF) |

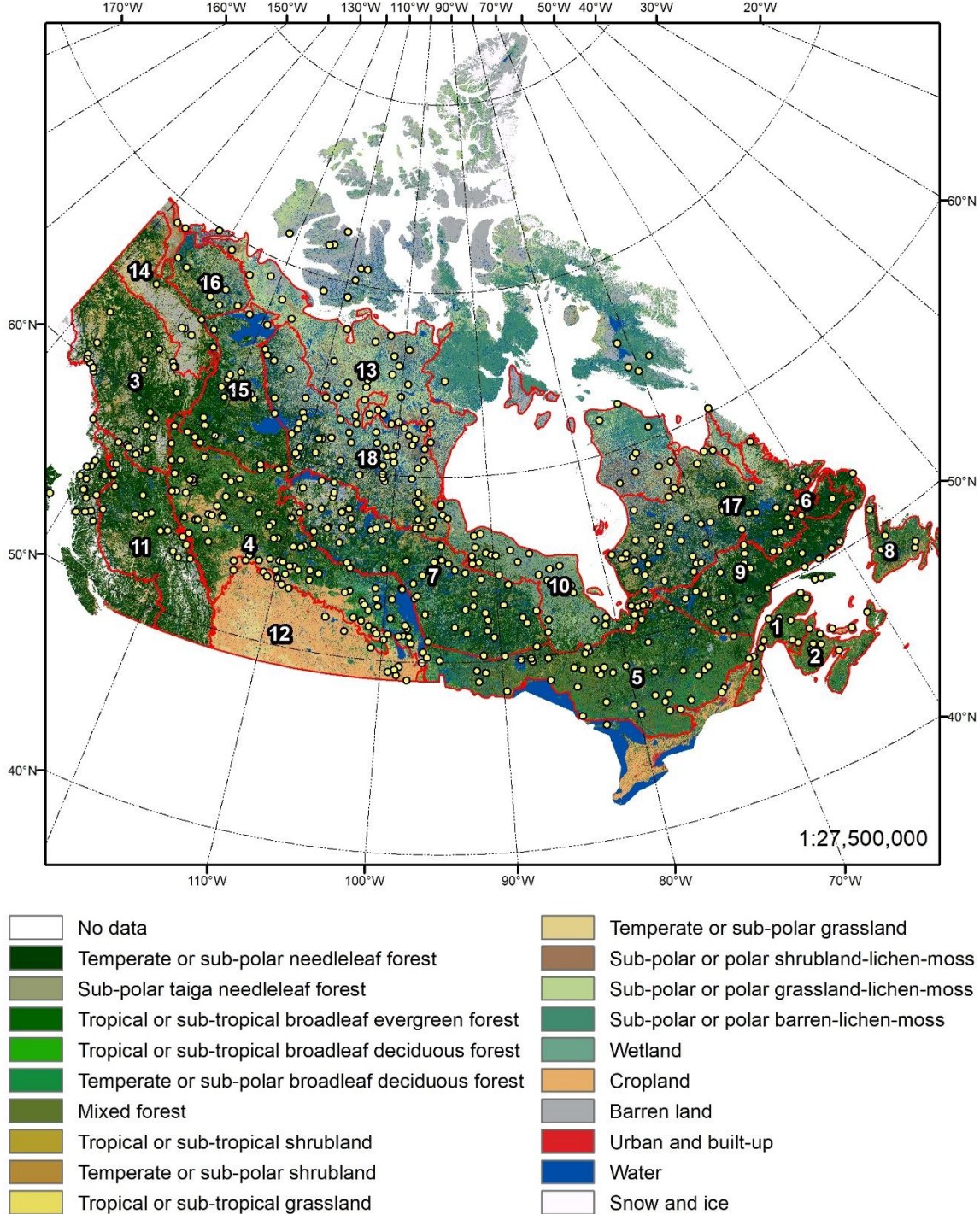

**Figure 1.** The 18 ecozones used in this study overlaid on the NALCMS (North America Land Change Monitoring System) land cover map of Canada. The yellow dots represent reference sample data sites used in this study.

## 2.2. Reference Sample Data

We used 567 randomly selected LC samples (Figure 1) classified by [40] through expert interpretation of Google Earth (GE) or Bing Maps (BM) high-resolution images and a false color composite from Landsat Enhanced Thematic Mapper Plus (ETM+) acquired in 2010. Available ground truth data from field campaigns were used to assist in interpretation of the samples. These sample data are representative at the 30m scale, and they are a subset of reference samples used to validate

the North America Land Change Monitoring System (NALCMS) LC dataset in [40]; thus they were classified using the same NALCMS legend. In this legend (see details below) treed wetland is not separated from herbaceous wetland, which is problematic given our focus is forested areas of Canada. To resolve this issue, the samples are reinterpreted using high-resolution images in GE/BM images focusing on samples classified as wetland. We reclassify the samples into forest and non-forest samples first, then use them as our reference to validate forest/non-forest pixels mapped by the high-resolution datasets used in this study.

*2.3. Regional LC Datasets Over Canada*

In recent years satellite data with finer resolutions (~25–30 m) have become readily available. This in combination with advanced computing power makes it possible to generate finer resolution LC products over large areas. Several LC products were generated using medium-to-high resolution optical satellite data over Canada. Recently a LC dataset for circa 2010 was generated at 30m resolution using Landsat images by the Canada Centre for Remote Sensing under the NALCMS program [40] (hereafter the NALCMS dataset). An innovative approach was used first to train and classify locally confined reference data over a large number of partially overlapping areas which ensured the optimization of the classifier to a local LC distribution; then a weighted combination of labels determined by the classifier in overlapping windows defined the final label for each pixel. The NALCMS dataset has 19 classes based on the United Nations Land Cover Classification System (LCCS, [41]) (Figure 1). Please note that four of these classes are not found in Canada (underlined in Table 1). An accuracy assessment based on 2011 randomly distributed samples showed that adjacent forest classes can be confused, for example coniferous with mixed forest, as well as deciduous forest, shrubland, shrub-covered wetlands, and certain croplands [40]. Confusion may also arise with the herbaceous class being misidentified as either low biomass croplands, or sparse coniferous forest along the northern tree line. Common to many remote sensing analyses, errors can be attributable to ambiguous spectral features and/or radiometric saturation occurring at leaf area index levels in the range of 3–5 [40,42], which generally apply to all satellite-derived LC products [19]. Nevertheless an overall accuracy of 76.6% was achieved, demonstrating quality and reliability as a land cover product representing Canada. The NALCMS dataset is used as a reference in our comparisons and assessments.

To better monitor and manage Canada's forests, the Earth Observation for Sustainable Development of Forests (EOSD) LC project was initiated as a partnership between the Canadian Forest Service and the Canadian Space Agency, with provincial and territorial participation and support. Based on the National Forest Inventory hierarchical classification system, the EOSD project produced a 23 class LC map of the forested area of Canada representing circa year 2000 conditions at 25 m resolution [43]. Over 480 Landsat-7 ETM+ images were classified-based upon an unsupervised hyperclustering, cluster merging and labeling method. Following release, EOSD was the most detailed and comprehensive map of the forested area of Canada. A unique feature of this dataset is that it provides detailed sub-classes for all vegetative LC classes (Table 1).

The third regional LC dataset was derived from the Moderate Resolution Imaging Spectroradiometer (MODIS) at 250 m resolution, and is available for 2000–2011 [44]. The development of temporally consistent LC time series from satellite data has proven difficult because multi-year observations are usually acquired under different conditions [10,35]. A change-based update approach was developed and applied to annual MODIS time series to generate a consistent LC dataset. The 2005 MODIS LC dataset for Canada [45] was used as the base map. The approach is similar to that employed to generate the annual CCI LC time series. However, the MODIS-derived dataset focuses on Canada only and benefitted from local (Canadian) expertise. It uses the same 19-class legend as the NALCMS dataset (Table 1). The MODIS LC maps for 2000 and 2010 are used as a reference for LC change detection between 2000 and 2010.

### 2.4. Global LC Datasets

The GLC2000 dataset was generated from SPOT-VEG data collected from November 1999 to December 2000 at 1 km resolution [9]. It was produced by 21 separate regional expert groups using an unsupervised classification method. Based on the LCCS, the regional products were merged to one global product with a generalized LCCS legend of 22 classes (Table 1). Assessment based on random sampling of reference sites globally estimated an accuracy of 68.6% [17].

Based on broad user consultations and lessons learned from the development of previous LC products, especially the GlobCover products, a unique LC mapping approach was developed to generate annual CCI LC datasets at 300 m for the period 1992–2015 [12,35,46]. This approach was developed and tested previously in Canada by [44,47]. There are three steps in the classification chain. First, a baseline map was generated using a combination of machine learning and unsupervised classification methods from the entire archive of ENVISAT/MERIS for 2003–2012. The reference database consisted of global, regional and local reference LC maps selected as the most accurate and thematically compatible ones for a region. In the classification algorithm, the whole globe was stratified into 22 equal-reasoning areas based on climatic and remote sensing conditions to account for local LC characteristics [48]. Then annual LC changes were detected at 1 km resolution from the AVHRR time series between 1992 and 1999, SPOT-VEG time series between 1999 and 2013 and PROBA-V time series between 2013 and 2015. The last step consists of backward or forward updating from the baseline map to produce the annual LC maps. This approach is designed to avoid the problem of significant year-to-year variations in LC labels not associated with actual LC changes which were found in the MODIS and GlobCover products [10,35], and enables the internal consistency of the annual LC time series, a major request from the climate modelling community. Based on the LCCS legend, the CCI LC maps have 22 level 1 classes, and 15 level 2 sub-classes (Table 1). Assessment based on the GlobCover validation database estimated an overall accuracy of 71% [12]. The CCI LC product for 2000 and 2010 are assessed relative to GLC2000 and finer resolution LC datasets. Given our reference dataset (NALCMS) is for 2010, we use the CCI dataset for 2010 in the following analyses unless otherwise specified.

### 2.5. Tree Cover Fraction Data

Given the large impact of forest cover on the simulation of winter albedo, we also include a tree cover fraction and forest cover change dataset produced by Hansen et al. [49] (hereafter the Hansen dataset). It was based on Landsat images at 30 m resolution. In contrast to the discrete LC classification datasets described above, the Hansen dataset is a vegetation continuous field product, in which the satellite spectral information was used to estimate the tree cover fraction in each pixel using a regression tree algorithm [50,51]. This may better represent heterogeneous areas than is possible by discrete LC classification. In the algorithm, Quickbird images (2 m resolution) were used to map tree crowns to relate tree cover fraction to spectral signals from Landsat. Tree cover was defined as canopy closure for all vegetation taller than 5 m in height. Forests are generally defined as taller than 3 m in the regional and global LC datasets. The different definitions in tree heights should not result in much difference in areas with mature forests, such as most boreal forests over Canada.

### 3. Methods

We use multiple methods to assess the above datasets at different scales. First, we reclassify the three high-resolution datasets (NALCMS, EOSD, and Hansen) into forest and non-forest categories, and evaluate their capability for discriminating forest cover from low vegetation and barren land using the reference sample data described in Section 2.2 at a 30 m scale. This provides some measure of the uncertainties in these datasets. Second, we transform all the LC datasets into a common legend and grid at 1 km resolution, compare the frequency of LC classes of all the LC datasets, and examine the producer and user accuracies of the GLC2000 and CCI datasets. Then we compute the sub-pixel

fractional accuracy of the CCI dataset using the 30 m NALCMS dataset as a reference. Finally, we convert the GLC2000 and CCI datasets to the four CLASS PFTs at 0.5 degree and compare the spatial agreement of the PFT maps.

### 3.1. Reclassify to Forest and Non-Forest at 30 m Scale

Through investigating the various class descriptions, it is straightforward to classify the reference sample data (Section 2.2), as well as the NALCMS and EOSD products into forest and non-forest categories. For instance, to obtain the forest category, we simply combine all the forest classes, which are LC 1-6 for reference samples and NALCMS, and LC81 and 211–233 for EOSD (Table 1), and the remaining classes are all combined as the non-forest category. The definition for wetland is different in the EOSD and the NALCMS maps (Table 1). There are three sub-classes for wetland in EOSD: wetland-treed (81), wetland-shrub (82), and wetland-herb (83). Although wetlands are largely herbaceous and shrubs in the NALCMS legend for Canada according to [40], there are trees in some wetland pixels in NALCMS according to the high-resolution GE/BM images. An approximation of wetland-treed pixels in NALCMS is derived based on the ratio of wetland-treed pixels and total wetland pixels in EOSD for each ecozone (Table 2). It is assumed that forested wetland occupies the same fraction of total wetland as in EOSD and this is added to the forest category in NALCMS to produce Adj-NALCMS.

Given that the NALCMS is for 2010 and the Hansen tree fraction data are for 2000, we apply the forest loss and gain information from the Hansen dataset [49] to the tree fraction data for 2000 to obtain an estimate of tree cover fraction for 2010 (hereafter Hansen2010). Different thresholds of tree cover fraction (from 5% to 25% with 5% increments) are used to classify the Hansen datasets into forest and non-forest categories. The results show that the 25% threshold provides the best match with the reference sample data and is therefore used in our analysis; the 25% tree cover threshold was also used in [51] to define forest classes.

We choose pixels closest to the locations of the reference sample points (Figure 1) from the original grid of the NALCMS, EOSD and Hansen datasets (their resolutions are all around 30 m), and compare the number of forest and non-forest samples correctly identified by each dataset. In addition, we reproject/regrid the forest/non-forest maps from the three datasets into a 300 m resolution common grid and compare the frequency of forest cover in the 18 ecozones across Canada.

### 3.2. Transform to the Common Legend

The different datasets cannot be compared directly because, as will be explained, they have different resolutions and different numbers of LC classes (Table 1). As in previous LC dataset comparison studies e.g., [16], we reproject/regrid all the datasets into a common grid and convert each dataset into a common legend. During the transformation process, we visually examine the spatial distribution of the LC classes from each of the datasets, making sure that LC classes converted into the same categories in the common legend are located in similar regions/ecozones.

#### 3.2.1. Reproject to the Common Grid and Convert to the Common Legend

When the GLC2000 dataset was involved in the analysis, we chose the associated latitude-longitude projection as our common grid since it has the lowest resolution. Pixel resolution in GLC2000 is 1/112°, which corresponds to 1 km at the equator. In the reprojection/regridding process, we kept track of the most and the second most abundant categories and the respective fractions in order to define pixels that meet the definition of the mosaic LC class (see details below). All other pixels were assigned the LC class corresponding with the most abundant class. When the GLC2000 dataset was not involved, all datasets with resolutions finer than 1 km were reprojected/regridded to a 300 m (0.0028°) common grid using the same approach.

Based on the characteristics of LC types over Canada, 13 classes were chosen for the common legend based on LCCS (Table 3); 12 of the classes are similar to the level I legend in the NALCMS

dataset [40]. We add a mosaic class of tree and other vegetation because this is a common LC category at 1 km resolution. Except for the EOSD dataset, the LC classes in all the datasets were fully developed in or comply with the LCCS legend. The 23 classes in EOSD can be easily merged to the corresponding broad categories in the common legend (e.g., combine classes of conifer-dense, conifer-open, and conifer-sparse into needleleaf forest (NF)) (Table 3). Although the forest definition in the EOSD legend looks different compared to the other legends, the sparse, open, and dense forest categories in EOSD can be transformed to corresponding classes in the LCCS legend (Table 4 in [52]). The forest definitions are also slightly different for GLC2000/CCI (>15% cover) and NALCMS/MODIS (>5% cover for sub-polar taiga needleleaf forest and >20% cover for other forest types). They all comply with the height requirement of above 3 m as in the LCCS legend for forest definition. Please note that LC classes are defined hierarchically by the height of the canopy layer ranging from trees to shrubs to herbaceous cover.

**Table 3.** The common legend and the merging rules for each land cover dataset.

| Common Legend | NALCMS/ MODIS | EOSD | CCI | GLC2000 |
|---|---|---|---|---|
| 1. Needleleaf forest (NF) | 1,2 | 211,212,213 | 70,71,72,80,81,82 | 4,5 |
| 2. Broadleaf forest (BF) | 5 | 221,222,223 | 50,60,61,62 | 1,2,3 |
| 3. Mixed forest (MF) | 6 | 231,232,233 | 90 | 6 |
| 4. Mosaic forest/other | | | 100,110 | 9 |
| 5. Shrubs | 8 | 51,52 | 120,121,122 | 11,12,10 |
| 6. Grassland | 10 | 100,110 | 130 | 13 |
| 7. Sparse Veg | 11,12,13 | 40 | 140,150,151,152,153 | 14 |
| 8. Wetland | 14 | 81,82,83 | 160,170,180 | 7,8,15 |
| 9. Cropland | 15 | N/A | 10,11,12,20,30,40 | 16, 17, 18 |
| 10. Barren land | 16 | 32,33 | 200,201,202 | 19 |
| 11. Urban and built-up | 17 | 34 | 190 | 22 |
| 12. Water | 18 | 20 | 210 | 20 |
| 13. Snow and ice | 19 | 31 | 220 | 21 |

**Table 4.** Cross-walking table for converting CCI LC classes into CLASS PFTs.

| ID | ESA-CCI Legend Description | NF | BF | Crop | Grass | Urban | Lake | Ocean | Bare |
|---|---|---|---|---|---|---|---|---|---|
| 10 | Cropland, rainfed (CR) | | | 1.0 | | | | | |
| 11 | CR Herbaceous cover | | | 1.0 | | | | | |
| 12 | CR Tree or shrub cover | | 0.5 | 0.5 | | | | | |
| 20 | Cropland, irrigated or post-flood | | | 1.0 | | | | | |
| 30 | Mosaic cropland (>50%)/natural vegetation (tree, shrub, herb) | 0.05 | 0.2 | 0.6 | 0.15 | | | | |
| 40 | Mosaic natural vegetation (tree, shrub, herb) >50%/crop | 0.075 | 0.275 | 0.4 | 0.25 | | | | |
| 50 | Tree cover broadleaved evergreen closed to open | | 1.0 | | | | | | |
| 60 | Tree cover broadleaved deciduous closed to open | | 0.85 | | 0.15 | | | | |
| 61 | Tree cover broadleaved deciduous closed | | 0.85 | | 0.15 | | | | |
| 62 | Tree cover broadleaved deciduous open | | 0.55 | | 0.35 | | | | 0.1 |
| 70 | Tree cover needleleaf evergreen closed to open | 0.75 | 0.1 | | 0.15 | | | | |
| 71 | Tree cover needleleaf evergreen, closed | 0.75 | 0.1 | | 0.15 | | | | |
| 72 | Tree cover needleleaf evergreen open | 0.35 | 0.05 | | 0.3 | | | | 0.3 |
| 80 | Tree cover needleleaf deciduous closed to open | 0.75 | 0.1 | | 0.15 | | | | |
| 81 | Tree cover needleleaf deciduous closed | 0.75 | 0.1 | | 0.15 | | | | |
| 82 | Tree cover needleleaf deciduous open | 0.35 | 0.05 | | 0.3 | | | | 0.3 |
| 90 | Tree cover Mixed | 0.35 | 0.4 | | 0.15 | | | | 0.1 |
| 100 | Mosaic tree and shrub (>50%)/herbaceous cover (<50%) | 0.15 | 0.45 | | 0.4 | | | | |
| 110 | Mosaic herbaceous cover (>50%)/tree and shrub (<50%) | 0.1 | 0.3 | | 0.6 | | | | |
| 120 | Shrubland | 0.2 | 0.4 | | 0.2 | | | | 0.2 |

**Table 4.** *Cont.*

| ID | ESA-CCI Legend Description | NF | BF | Crop | Grass | Urban | Lake | Ocean | Bare |
|---|---|---|---|---|---|---|---|---|---|
| 121 | Shrubland evergreen | 0.3 | 0.3 | | 0.2 | | | | 0.2 |
| 122 | Shrubland deciduous | | 0.6 | | 0.2 | | | | 0.2 |
| 130 | Grassland | | | | 0.6 | | | | 0.4 |
| 140 | Lichens and mosses | | | | 0.6 | | | | 0.4 |
| 150 | Sparse vegetation (tree, shrub, herb) <15% | 0.02 | 0.08 | | 0.05 | | | | 0.85 |
| 151 | Sparse tree (<15%) | 0.02 | 0.08 | | 0.05 | | | | 0.85 |
| 152 | Sparse shrub (<15%) | 0.02 | 0.08 | | 0.05 | | | | 0.85 |
| 153 | Sparse herbaceous cover (<15%) | | | | 0.15 | | | | 0.85 |
| 160 | Tree cover, flooded fresh/brackish | | 0.6 | | 0.2 | | 0.2 | | |
| 170 | Tree cover, flooded saline water | | 0.8 | | | | | 0.2 | |
| 180 | Shrub or herbaceous cover, flooded | 0.15 | 0.15 | | 0.4 | | 0.15 | 0.15 | |
| 190 | Urban areas | 0.025 | 0.025 | | 0.15 | 0.75 | 0.05 | | |
| 200 | Bare areas | | | | | | | | 1.0 |
| 201 | Consolidated bare areas | | | | | | | | 1.0 |
| 202 | Unconsolidated bare areas | | | | | | | | 1.0 |
| 210 | Water bodies | | | | | | 1.0 | | |
| 220 | Permanent snow and ice | | | | | | | | 1.0 |

Since treed wetland is not explicitly separated from herbaceous wetland in the NALCMS/MODIS legend, LC classes of tree cover flooded with fresh/saline water in the CCI and GLC2000 legends are labeled as wetland under the common legend (Table 3). While the mosaic classes in the two global datasets (CCI and GCL2000) can be linked directly to the mosaic category in the common legend, there are no mosaic categories in the regional datasets. We define the mosaic class for the regional datasets based on the following rules. If a pixel has (1) the most abundant type as shrub with fractional coverage > 40%, and the second most abundant type is from one of the tree types (e.g., 1,2,5,6 for NALCMS) with fractional coverage > 10%; or (2) has the most abundant type as grass with fractional coverage > 50%, and the second most abundant type is from one of the trees or shrubs (e.g., 1,2,5,6,8 for NALCMS) with fractional coverage > 10%, it is labeled as mosaic forest and other vegetation. This is similar to rules used to define the mosaic classes in the CCI dataset (with labels of 100 and 110). If none of the conditions are met, the pixel remains as a shrub or grass class.

### 3.2.2. Comparison of LC Datasets at 1 km Scale

We compare the spatial distribution of the LC classes and the number of pixels for each class under the common legend over Canada. Using NALCMS as a reference, we compute the producer and user accuracies for the CCI and GLC2000 datasets. The producer's accuracy is the complement of the omission error [53], and it is the percentage of pixels of a given LC class in the reference dataset (in this case NALCMS) which are correctly identified in the LC dataset being assessed. The user's accuracy is the complement of the commission error, and it is the percentage of pixels in a given LC class in the dataset being assessed that are correctly identified with respect to the reference dataset. Pflugmacher et al. [54] showed that the choice of sampling unit significantly affected accuracy estimates. When the finer resolution datasets (i.e., CCI at 300 m) were regridded into the same grid of the coarser resolution dataset (i.e., GLC2000 at 1 km), the finer resolution datasets performed better in terms of overall accuracy. This is because the finer resolution datasets tend to have more homogenous pixels than the coarser resolution datasets. However, this advantage is reduced for larger pixel blocks [54]. Therefore, we compute the producer/user accuracies for 10 km by 10 km pixel blocks. In addition, we compare the frequency of LC classes from the CCI and GLC2000 datasets with that from the NALCMS dataset for the 18 ecozones across Canada, with a focus on the forest LC classes.

### 3.3. Sub-Pixel Fractional Accuracy at 300 m Scale

Sub-pixel fractional error matrices were introduced by Latifovic and Olthof [16]. They are produced by assigning sub-dominant LC classes from all fine-resolution pixels in the reference data to

the corresponding single coarse-resolution pixel. This allows a quantitative assessment of the fractional composition of each class in the coarse-resolution map, which will be useful for the partitioning of the coarse-resolution dataset into PFTs. In this study, the NALCMS data at 30 m resolution is used to compute the sub-pixel fractional error matrices of the CCI dataset based on the 13-class common legend. This is first done for each ecozone, then the mean error matrices are computed from the 18 ecozones across Canada.

Previous studies showed that landscape heterogeneity has a large impact on LC mapping accuracy [16,55,56]. To quantify landscape heterogeneity, 3x3 pixel neighborhoods were assessed for the CCI dataset following [19]. A neighborhood is considered homogenous if only one LC class is present. For comparison, we also compute the sub-pixel fractional error matrices for homogenous areas.

### 3.4. PFT Mapping at 0.5 Degree Scale

CLASS requires grid-averaged parameters for LC related variables, such as surface albedo, leaf area index, surface roughness and rooting depth values. These are computed as weighted averages of typical PFT-level parameter values and LC fractions from GLC2000 using the cross-walking tables. Based on expert knowledge of global biomes and class descriptions, Wang et al. [57] created a cross-walking table to convert the GLC2000 22 LC classes to nine vegetation PFTs for use in CTEM, which were combined onto the four broad PFTs recognized by CLASS (Table 5).

**Table 5.** Cross-walking table for converting GLC2000 LC classes into CLASS PFTs.

| GLC2000 Legend Description | NF | BF | Crop | Grass | Urban | Lake | Ocean | Bare |
|---|---|---|---|---|---|---|---|---|
| 1 – Tree cover, broadleaved, evergreen | | 1.0 | | | | | | |
| 2 – Tree cover, broadleaved, deciduous, closed | | 1.0 | | | | | | |
| 3 – Tree cover, broadleaved, deciduous, open | | 0.6 | | 0.2 | | 0.1 | | 0.1 |
| 4 – Tree cover, needleleaved, evergreen | 1.0 | | | | | | | |
| 5 – Tree cover, needleleaved, deciduous | 0.8 | | | 0.1 | | | | 0.1 |
| 6 – Tree cover, mixed leaf type | 0.4 | 0.5 | | 0.1 | | | | |
| 7 – Tree cover, regularly flooded, fresh water | | 0.5 | | | | 0.5 | | |
| 8 – Tree cover, regularly flooded, saline water | | 0.5 | | | | | 0.5 | |
| 9 – Mosaic: tree cover / other natural vegetation | | 0.6 | | 0.2 | | | | 0.2 |
| 10 – Tree cover, burnt | 0.2 | 0.2 | | 0.3 | | | | 0.3 |
| 11 – Shrub cover, closed-open, evergreen | | 0.6 | | 0.2 | | 0.1 | | 0.1 |
| 12 – Shrub cover, closed-open, deciduous | | 0.4 | | 0.3 | | | | 0.3 |
| 13 – Herbaceous cover, closed-open | | | | 0.7 | | | | 0.3 |
| 14 – Sparse herbaceous or sparse shrub cover | | 0.1 | | 0.1 | | | | 0.8 |
| 15 – Regularly flooded shrub and/or herbaceous cover | | 0.5 | | 0.3 | | 0.1 | | 0.1 |
| 16 – Cultivated and managed areas | | | 0.5 | 0.4 | | | | 0.1 |
| 17 – Mosaic: cropland / tree cover / other natural veg | | 0.2 | 0.5 | 0.2 | | | | 0.1 |
| 18 – Mosaic: cropland / shrub and/or grass cover | | 0.1 | 0.5 | 0.3 | | | | 0.1 |
| 19 – Bare areas | | | | | | | | 1.0 |
| 20 – Water bodies | | | | | | 1.0 | | |
| 21 – Snow and ice | | | | | | | | 1.0 |
| 22 – Artificial surfaces and associated areas | | | | | 1.0 | | | |

The CCI LC product user guide [12] provides a cross-walking table for converting the 37 CCI LC classes into ten PFTs. This table was originally developed by [58] based on recommendations from experts in the remote sensing and climate modelling communities. It includes four tree PFTs, four shrub PFTs and two grass PFTs. CLASS does not yet have explicit shrub PFTs (research on including shrubs as a separate PFT is ongoing), so the four shrub PFTs were merged into either the NF or BF tree PFTs as was done in creating the GLC2000 table (Tables 4 and 5). Based on Tables 4 and 5, the

four CLASS PFTs are generated from GLC2000 and CCI (for 2000) datasets respectively on a $0.5 \times 0.5$ degree latitude/longitude grid for a sub-domain of North America.

## 4. Results

### 4.1. Comparison of Forest Cover

First, we compare the NALCMS, Hansen, and EOSD datasets with the reference sample data at 30 m resolution for selected ecozones, in which there are relatively abundant sample data. Table 6a,b present the fraction of samples identified correctly by the NALCMS, EOSD, and Hansen datasets for the forest and non-forest categories respectively in nine ecozones. The results show that NALCMS and EOSD perform similarly for classifying the forest (0.83 and 0.70) and non-forest (0.87 and 0.76) samples. In contrast, the Hansen dataset has a higher correct rate for classifying the forest (0.83) than the non-forest (0.69) samples. For the Hansen dataset, there are large differences in the correct rates for forest vs. non-forest especially in EZ7 (0.94 vs. 0.33) and EZ10 (0.84 vs. 0.57), both with a relatively large percentage of wetland (Table 2). This suggests that the Hansen dataset may overestimate tree cover (at least for tree cover less than 25%) in wetland environments, which we confirmed by examining high-resolution images in GE/BM. Overall NALCMS performs the best among the three datasets.

**Table 6.** (**a**) The fraction of forest and samples correctly identified by NALCMS, EOSD and Hansen2010 (adjusted using forest loss/gain data from Hansen2000); (**b**) The fraction of non-forest samples correctly identified by NALCMS, EOSD and Hansen2010 (adjusted using forest loss/gain data from Hansen2000).

| (a) Ecozone | NALCMS | EOSD | Hansen | Number of Forest Samples |
|---|---|---|---|---|
| 3-Boreal Cordillera | 0.71 | 0.43 | 0.86 | 7 |
| 4-Boreal Plains | 0.91 | 0.77 | 0.94 | 35 |
| 7-Boreal Shield | 0.71 | 0.74 | 0.94 | 31 |
| 9-Boreal Shield | 1.00 | 0.73 | 0.82 | 11 |
| 10-Hudson Plains | 0.79 | 0.68 | 0.84 | 19 |
| 11-Montane Cordillera | 1.00 | 0.80 | 0.80 | 5 |
| 15-Taiga Plains | 0.89 | 0.67 | 0.67 | 9 |
| 17-Taiga Shield | 0.84 | 0.64 | 0.72 | 25 |
| 18-Taiga Shield | 0.69 | 0.69 | 0.62 | 13 |
| Mean/Total | **0.83** | **0.70** | **0.83** | **155** |
| (b) Ecozone | NALCMS | EOSD | Hansen | Number of Non-Forest Samples |
| 3-Boreal Cordillera | 0.95 | 1.00 | 0.84 | 19 |
| 4-Boreal Plains | 0.82 | 0.70 | 0.60 | 50 |
| 7-Boreal Shield | 0.88 | 0.67 | 0.33 | 24 |
| 9-Boreal Shield | 0.77 | 0.69 | 0.92 | 13 |
| 10-Hudson Plains | 0.93 | 0.57 | 0.57 | 14 |
| 11-Montane Cordillera | 0.87 | 0.67 | 0.87 | 15 |
| 15-Taiga Plains | 0.80 | 0.80 | 0.60 | 10 |
| 17-Taiga Shield | 0.81 | 0.81 | 0.84 | 31 |
| 18-Taiga Shield | 0.98 | 0.85 | 0.74 | 47 |
| Mean/Total | **0.87** | **0.76** | **0.69** | **223** |

The proportions of forest cover from the above three datasets are obtained for each ecozone by aggregating the ~30 m data into a 300 m common grid (Figure 2). The forest cover from NALCMS agrees well with those from Hansen2010 (within 10%) in most regions except for EZ7, 10, and 16. Considering the completely different methods used to generate the datasets, and the different definitions for forest cover, the 10% difference is understandable. There are relatively large percentages of wetland in EZ7 (11%) and EZ10 (37%) in NALCMS (Figure 2 and Table 2). The agreement improves for these two ecozones after wetland-treed pixels are added to the forest cover as shown in Adj_NALCMS (Figure 2). EZ16 only has a small percentage of wetland (2%), but it has a large fraction of taiga NF (0.49, Table 2), which is defined with a cover of greater than 5% (and an upper limit of ~25%). These taiga NF areas would not have been mapped by Hansen given the use of a 25% tree cover fraction threshold for forest

definition, likely explaining the much higher forest cover in NALCMS than Hansen in EZ16 (Figure 2). EZ10 and EZ15–18 all have a relatively large fraction of taiga NF (> 0.15, Table 2), which may to some extent contribute to the differences in forest cover mapped by the two datasets. The proportion of forest cover from EOSD tends to be lower than that from Hansen and NALCMS in most ecozones across Canada.

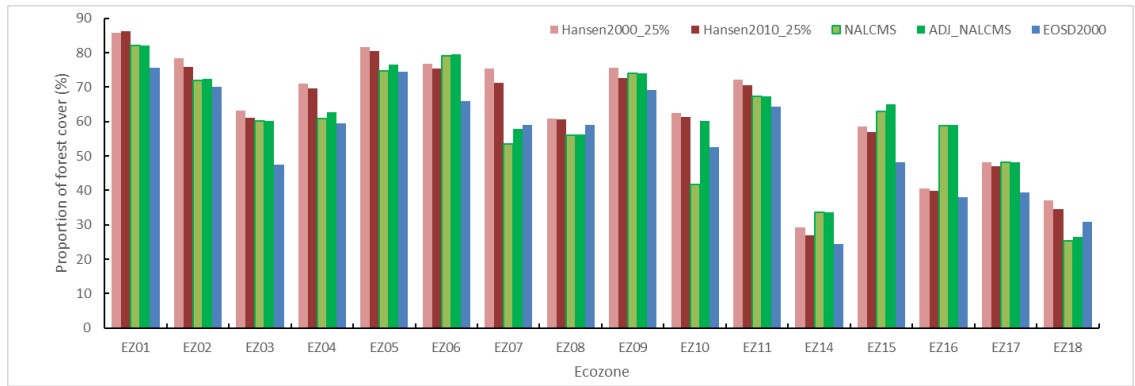

**Figure 2.** The proportion of forest cover from different datasets for 16 ecozones dominated with forest. Hansen2010 is obtained from Hansen2000 plus forest loss/gain during 2000-2010. ADJ_NALCMS is the same as NALCMS except that it includes wetland-treed pixels. See text for details.

These results suggest that there are relatively large uncertainties in forest LC mapping in EZ7 and 10 due to larger fractions of wetland in NALCMS. Though the fraction of treed wetland can be estimated based on information from EOSD as above, EOSD mapped a much larger area of wetland than NALCMS (see below), plus the type of forest is not identified/given. We therefore do not attempt to separate treed wetland from herbaceous wetland but keep this in mind in the following analyses. The sparse taiga NF in NALCMS is not found in the two global LC datasets, which will add some uncertainty to the results. The NF fractions from NALCMS should be biased high in ecozones with a high percentage of taiga NF (Table 2).

*4.2. Comparison of LC Datasets at 1 km Resolution*

4.2.1. The Spatial Patterns and Pixel Counts

The spatial distribution of LC classes from the NALCMS, MODIS and the CCI maps are similar, while there are relatively large differences from the EOSD and GLC2000 maps compared to the other maps (Figure 3). Relative to the NALCMS map (Figure 3a), the EOSD map (Figure 3b) shows less NF and mosaic classes but more shrub and wetland (Table 7 and Figure 4). EOSD mapped at least 100% more wetland than any other LC product, likely due to the different definition of wetland in its legend (as wetland-treed is a treed/forest category [52]). The MODIS maps have less NF but more MF category. The CCI maps have more NF and sparse vegetation but less area classified as barren lands. CCI mapped over one million more pixels of NF than all the other maps, while it mapped a bit less of all the other tree categories relative to NALCMS. The GLC2000 map has more mosaic, shrub, and sparse vegetation and less NF, wetland, and barren land. The mosaic dominated areas in GLC2000 are mainly located in NF dominated regions according to the other maps. GLC2000 has ~50% more snow and ice pixels than all the other maps. The large differences in sparse vegetation and barren lands may be partly due to the different definitions in the legends of each dataset. For example, barren lands in NALCMS/MODIS were defined as areas characterized by bare rock, gravel, sand, silt, clay, or other earthen material, with little or no "green" vegetation present and vegetation generally accounts for less than 10% of total cover, while in GLC2000, bare areas were defined as primarily non-vegetated areas containing less than four percent vegetation during at least 10 months a year, including areas like bare rock and sands.

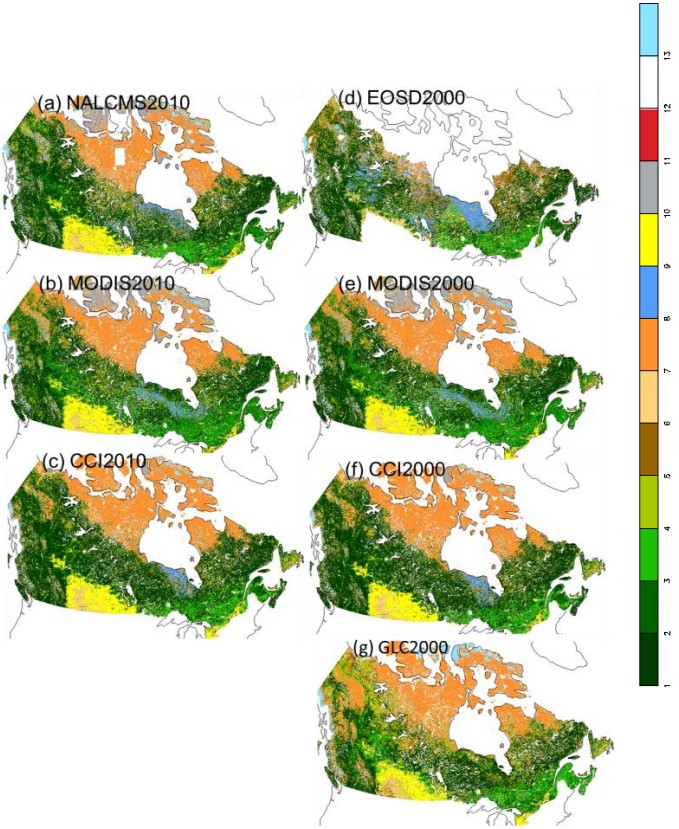

**Figure 3.** Maps of land cover under the common legend. The 13 classes are: 1-NF, 2-BF, 3-MF, 4-Mosaic, 5-Shrub, 6-Grass, 7-Sparse, 8-Wland, 9-Crop, 10-Barren, 11-Urban, 12-Water, 13-Snow (see Table 3 for details).

**Table 7.** Pixel counts (million) for each of the LC classes under the common legend at 1km over Canada. Please note that EOSD is only available for forested areas.

| LC Class | NF | BF | MF | Mosaic | Shrub | Grass | Sparse | Wland | Crop | Barren | Urban | Water | Snow |
|----------|------|------|------|--------|-------|-------|--------|-------|------|--------|-------|-------|------|
| NALCMS2010 | 5.63 | 0.60 | 0.99 | 0.74 | 0.50 | 0.40 | 4.33 | 0.59 | 0.91 | 1.96 | 0.03 | 2.11 | 0.25 |
| EOSD2000 | 4.44 | 0.51 | 0.95 | 0.08 | 1.74 | | | 1.22 | | | | | |
| MODIS2000 | 5.12 | 0.71 | 1.86 | 0.50 | 0.50 | 0.20 | 4.46 | 0.60 | 0.90 | 1.96 | 0.01 | 1.96 | 0.28 |
| MODIS2010 | 4.82 | 0.77 | 1.81 | 0.64 | 0.60 | 0.25 | 4.46 | 0.58 | 0.90 | 1.96 | 0.01 | 1.97 | 0.28 |
| CCI2000 | 6.73 | 0.42 | 0.80 | 0.58 | 0.58 | 0.43 | 5.67 | 0.39 | 0.92 | 0.37 | 0.01 | 1.96 | 0.20 |
| CCI2010 | 6.78 | 0.42 | 0.81 | 0.62 | 0.58 | 0.43 | 5.62 | 0.35 | 0.92 | 0.37 | 0.02 | 1.95 | 0.20 |
| GLC2000 | 4.31 | 0.35 | 1.16 | 1.87 | 1.33 | 0.23 | 6.09 | 0.06 | 0.78 | 0.00 | 0.01 | 2.18 | 0.68 |

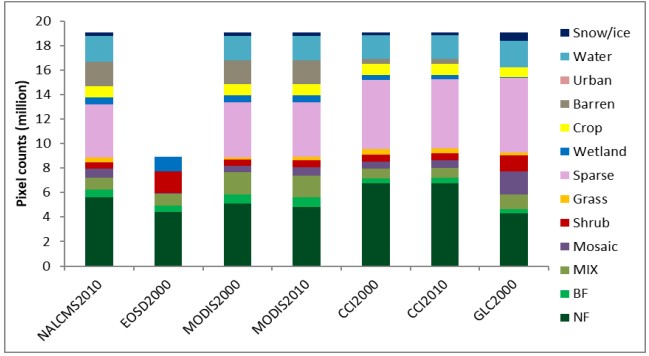

**Figure 4.** Pixel counts for each land cover class under the common legend across Canada. Please note that the EOSD dataset is only for forested areas.

4.2.2. Changes in LC Classes between 2000 and 2010

The MODIS and CCI datasets are available for multiple years with an overlap period of 2000-2011. We include both the 2000 and 2010 datasets in our analyses and compare the pixel counts of the LC classes from the two datasets. In addition, changes in forest cover (loss/gain) are available from [47] for the period 2000–2012. Thus, we also include a comparison of changes in forest cover from MODIS, CCI, and Hansen between 2000 and 2010 for ecozones dominated by forests. For MODIS and CCI, forest cover fractions are the ratio of the total number of forested pixels (LC1-6 for MODIS and LC50-100, LC160 and LC170 for CCI) and the total number of pixels at the 300 m common grid in each ecozone.

Based on the pixel counts in 2000 and 2010, overall the MODIS maps show larger changes than the CCI maps during the period, and the change direction is not always consistent between the two datasets (Table 7). For example, MODIS shows a 0.3 million decrease in NF from 2000 to 2010, while CCI shows a 0.05 million increase in NF during the same period. For the shrub class, MODIS shows an increase of 0.1 million, while there is no change based on CCI. From 2000 to 2010, the total number of pixels with a forest type (LC1-4) decreased from 8.19 to 8.04 million according to MODIS, while it increased from 8.37 to 8.47 million according to CCI.

Figure 5 shows changes in forest cover in each ecozone between 2000 and 2010. In general, both the Hansen and the MODIS datasets show a decrease (negative value) of forest cover from 2000 to 2010, with the largest decrease in EZ7, while CCI shows the opposite sign of change (forest gain) in most regions during the same period. Pouliot et al. [44] suggested that the decreases in NF were attributable mostly to fire, insect damage, and some harvesting, which was likely not detected by CCI. The CCI user manual [12] suggested that small changes before 2003 may not be captured by the CCI maps because change detection was based on coarse-resolution satellite data. This will be discussed further in the discussion section.

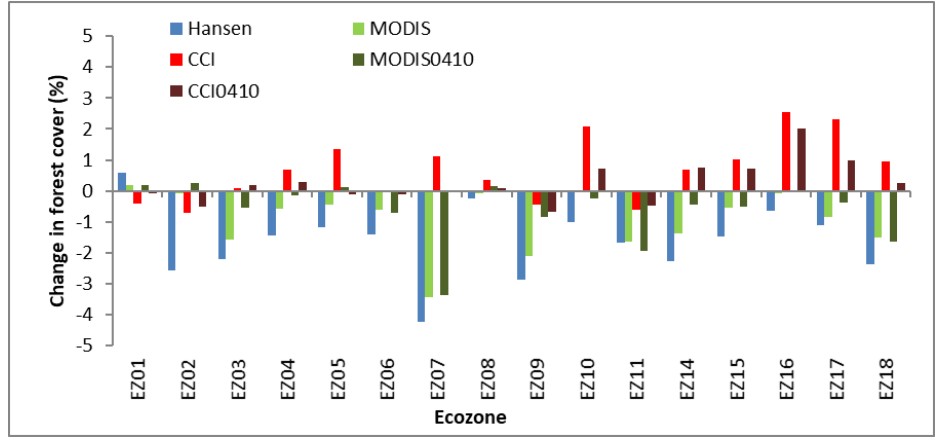

**Figure 5.** Change in forest cover from the Hansen, MODIS, and CCI datasets between 2000 and 2010 (solid lines). The dashed lines represent changes between 2004 and 2010. The positive values represent forest gain in 2010 and vice versa.

4.2.3. The Producer and User Accuracies

Figure 3 and Table 7 show that NF is the dominant forest type in Canada, so here we only present the accuracy results for NF. For the CCI dataset, the producer accuracies are generally high (dark red, > 85%) across Canada except for the Hudson Plains and some high relief areas in northwestern Canada (Figure 6a). In contrast, the user accuracies are in general lower than the producer accuracies except for some areas in Quebec and small areas in western Canada, with values less than 50% (blue to green color in Figure 6b) in large areas, such as the taiga shield in Quebec and areas west of Hudson Bay. Areas with high producer but low user accuracies indicate an over-mapping of NF by CCI. For the GLC2000 dataset, the producer accuracies are generally low across Canada but the user accuracies

are relatively higher in most areas of Canada, which indicates an under-mapping of NF (Figure 6c,d). Areas to the southwest of Hudson Bay are an exception where there are high producer but low user accuracies in GLC2000, indicating an over-mapping of NF there. As mentioned in Section 4.1, there are relatively large uncertainties in forest mapping in EZ7 and 10, due to large fractions of wetland in NALCMS. Thus, there are relatively large uncertainties in the accuracy estimation in those regions.

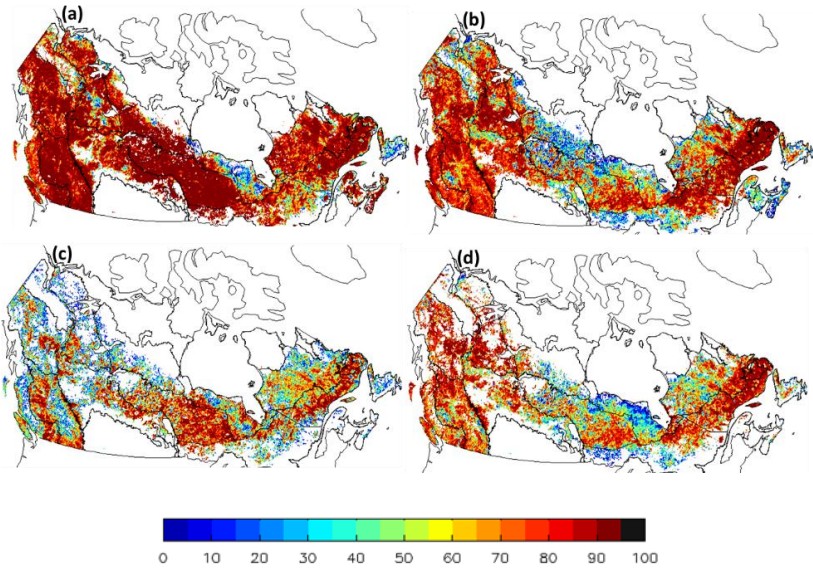

**Figure 6.** Producer (**a**,**c**) and user (**b**,**d**) accuracy (%) for needleleaf forest in the CCI (top) and GLC2000 (bottom) datasets relative to NALCMS at 10 km by 10 km pixel blocks.

The frequencies of NF, MF, and Mosaic classes for ecozones with more than 10% cover are shown in Figure 7. The results show that relative to NALCMS, the GLC2000 dataset mapped less NF for most ecozones except for EZ7 and 10, while the CCI dataset mapped more or nearly the same NF (Figure 7a). These are consistent with the accuracy assessment results shown in Figure 6. If we assume all wetland-treed pixels are NF and add that to NALCMS (Table 2), this mainly increases the percentage of NF in EZ7 and 10. The GLC2000 dataset mapped more MF and Mosaic classes than NALCMS and CCI in most ecozones (Figure 7b,c).

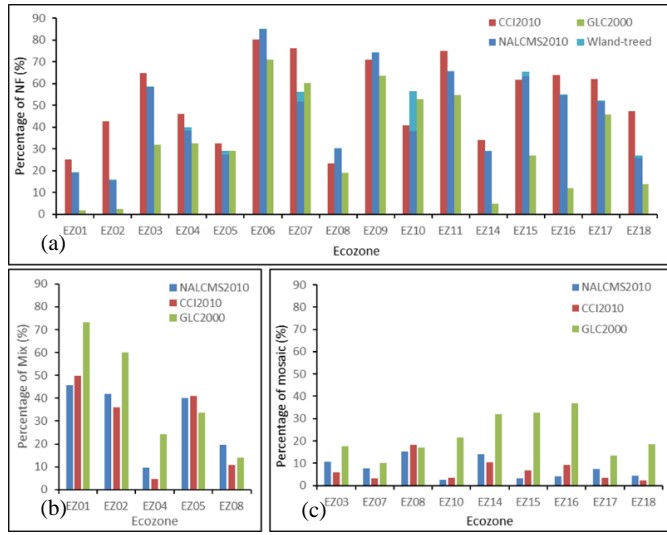

**Figure 7.** The proportion of needleleaf forest (**a**), mixed forest (**b**) and Mosaic of tree and other vegetation (**c**) in ecozones with greater than 10% cover (at least one dataset) from each datasets.

### 4.3. Sub-Pixel Fractional Error Matrices of CCI

Table 8 shows the fractional error matrices revealing more detailed class compositions of the CCI map. Values at the rows show the mean sub-pixel fractional cover of LC classes in the NALCMS dataset contributing to a given class in the CCI dataset across 18 ecozones in Canada. Values at the diagonal of the table are the sub-pixel fractional accuracies, which show the fraction of a given CCI class being mapped as that class by NALCMS. For example, the MF class in CCI is mainly composed of NF (0.23), BF (0.24), MF (0.28), Shrub (0.13) and other classes with small fractions in NALCMS. This suggests that near equal fractions should be assigned to NF and BF for the MF class in CCI in its CW table, which is the case in Table 5, while for the Mosaic class in CCI, nearly double the fraction should be assigned to NF (0.22) than to BF (0.12). However, the fractions for NF are 1/3 of BF in Table 5 for the two mosaic classes (100/110), likely impropriate for use in Canada. The accuracy for the wetland class is rather low (0.16), with most of the wetland class in CCI composed of NF (0.35) and shrub (0.12) in NALCMS. This suggests large discrepancies in this class between the two datasets.

**Table 8.** (**a**) The mean sub-fractional error matrix for CCI (rows) relative to NALCMS (columns) based on the 13-class common legend across 18 ecozones over Canada. Value of zero is omitted. Please note that there is no mosaic class in NALCMS. The 13LC classes are: 1-NF, 2-BF, 3-MF, 4-Mosaic, 5-Shrub, 6-Grass, 7-Sparse, 8-Wland, 9-Crop, 10-Barren, 11-Urban, 12-Water, 13-Snow; (**b**) Same as in (**a**) but for homogenous CCI pixels only.

**(a)**

| CLASS | 1 | 2 | 3 | 4 | 5 | 6 | 7 | 8 | 9 | 10 | 11 | 12 | 13 |
|---|---|---|---|---|---|---|---|---|---|---|---|---|---|
| 1 | **0.57** | 0.04 | 0.10 | | 0.05 | 0.04 | 0.06 | 0.05 | 0.01 | 0.01 | | 0.07 | |
| 2 | 0.10 | **0.37** | 0.17 | | 0.11 | 0.05 | 0.13 | 0.02 | 0.02 | 0.01 | 0.01 | 0.02 | |
| 3 | 0.23 | 0.24 | **0.28** | | 0.13 | 0.02 | 0.04 | 0.03 | 0.01 | | 0.01 | 0.02 | |
| 4 | 0.22 | 0.12 | 0.06 | | 0.24 | 0.06 | 0.11 | 0.08 | 0.03 | 0.01 | 0.03 | 0.04 | |
| 5 | 0.24 | 0.08 | 0.04 | | **0.25** | 0.10 | 0.12 | 0.07 | 0.02 | 0.03 | 0.02 | 0.02 | |
| 6 | 0.17 | 0.05 | 0.01 | | 0.15 | **0.15** | 0.21 | 0.06 | 0.06 | 0.08 | 0.02 | 0.03 | |
| 7 | 0.18 | 0.02 | 0.01 | | 0.11 | 0.12 | **0.22** | 0.08 | 0.01 | 0.17 | 0.01 | 0.05 | 0.01 |
| 8 | 0.35 | 0.07 | 0.07 | | 0.12 | 0.04 | 0.06 | **0.16** | 0.01 | 0.01 | 0.01 | 0.08 | |
| 9 | 0.06 | 0.15 | 0.05 | | 0.17 | 0.08 | | 0.01 | **0.37** | 0.04 | 0.05 | 0.01 | |
| 10 | 0.10 | 0.06 | 0.03 | | 0.07 | 0.12 | 0.15 | 0.04 | 0.01 | **0.26** | 0.04 | 0.07 | 0.01 |
| 11 | 0.04 | 0.04 | 0.02 | | 0.05 | 0.01 | 0.02 | 0.01 | 0.03 | 0.03 | **0.69** | 0.06 | |
| 12 | 0.07 | 0.01 | 0.01 | | 0.01 | 0.01 | 0.02 | 0.01 | | | | **0.74** | |
| 13 | | | | | | 0.01 | | | | 0.31 | | | **0.67** |
| Max | 0.91 | 0.61 | 0.53 | | 0.48 | 0.65 | 0.84 | 0.51 | 0.85 | 0.88 | 0.92 | 0.95 | 0.92 |

**(b)**

| CLASS | 1 | 2 | 3 | 4 | 5 | 6 | 7 | 8 | 9 | 10 | 11 | 12 | 13 |
|---|---|---|---|---|---|---|---|---|---|---|---|---|---|
| 1 | **0.71** | 0.03 | 0.08 | | 0.03 | 0.03 | 0.03 | 0.05 | 0.00 | 0.01 | 0.00 | 0.04 | |
| 2 | 0.07 | **0.47** | 0.15 | | 0.10 | 0.04 | 0.13 | 0.01 | 0.01 | 0.00 | 0.00 | 0.01 | |
| 3 | 0.17 | 0.29 | **0.35** | | 0.12 | 0.01 | 0.03 | 0.01 | 0.00 | 0.00 | 0.01 | 0.01 | |
| 4 | 0.21 | 0.14 | 0.06 | | 0.32 | 0.05 | 0.08 | 0.10 | 0.01 | 0.00 | 0.00 | 0.02 | 0.00 |
| 5 | 0.14 | 0.07 | 0.03 | | **0.39** | 0.09 | 0.15 | 0.06 | 0.03 | 0.03 | 0.00 | 0.01 | 0.00 |
| 6 | 0.15 | 0.04 | 0.01 | | 0.14 | **0.18** | 0.23 | 0.08 | 0.06 | 0.08 | 0.01 | 0.01 | 0.00 |
| 7 | 0.14 | 0.00 | 0.00 | | 0.08 | 0.13 | **0.26** | 0.07 | 0.01 | 0.22 | 0.02 | 0.07 | 0.01 |
| 8 | 0.26 | 0.05 | 0.05 | | 0.10 | 0.05 | 0.11 | **0.29** | 0.00 | 0.00 | 0.00 | 0.07 | |
| 9 | 0.03 | 0.14 | 0.03 | | 0.07 | 0.05 | 0.00 | 0.01 | **0.57** | 0.06 | 0.04 | 0.01 | 0.00 |
| 10 | 0.06 | 0.04 | 0.01 | | 0.07 | 0.14 | 0.15 | 0.03 | 0.00 | **0.35** | 0.04 | 0.05 | 0.02 |
| 11 | 0.00 | 0.01 | 0.00 | | 0.01 | 0.00 | 0.00 | 0.02 | 0.01 | 0.01 | **0.92** | 0.01 | |
| 12 | 0.01 | 0.00 | 0.00 | | 0.00 | 0.00 | 0.00 | 0.00 | | 0.00 | 0.00 | **0.80** | |
| 13 | 0.00 | | 0.00 | | 0.00 | 0.02 | 0.00 | 0.00 | | 0.23 | | 0.00 | **0.74** |
| Frac | 0.55 | 0.29 | 0.28 | | 0.1 | 0.31 | 0.54 | 0.21 | 0.79 | 0.24 | 0.23 | 0.43 | 0.41 |

Unlike most previous studies, which used limited sample points or sites in the assessments, we include all CCI pixels for a given class in each ecozone, and the ecozones are large (composed of millions of pixels each), and thus with low homogeneity (see last column in Table 2). NF is the dominant LC class for 15 out of the 18 ecozones, and the remaining LC classes do not cover large areas of Canada. These may partly explain the rather low accuracies in Table 8a, especially for LC3 to LC8 (<0.3). The bottom row in Table 8a shows the maximum accuracy for a given class among the 18 ecozones. The maximum accuracies for LC class of MF, shrub and wetland are still quite low (<0.6), which are likely due to the low mapping accuracies of these classes in both datasets [12,40]. The accuracy for LC classes of urban, water, and snow/ice are among the highest in Table 8a. These classes were largely mapped using external datasets in the CCI product [12]. There are some improvements in the accuracies for most classes when only homogeneous areas are considered (Table 8b). However, the fraction of each class occurring in homogeneous areas is low (< 0.31) except for LC class 1, 7, 9, 12–13 (last row in Table 8b), consistent with the low coverage of most LC classes in Canada (Tables 2 and 7).

### 4.4. Comparison of CLASS PFTs Derived from GLC2000 and CCI

There are large differences in the fractional coverage of all four PFTs from the GLC2000 and the CCI datasets (Figure 8). Compared to the CCI dataset, the GLC2000 dataset mapped more NF fractions in central Canada, the Hudson Plains, and southern Quebec, while it mapped smaller NF fractions in northwestern NA (Figure 8a). The GLC2000 dataset tends to map more BF fractions in areas where it mapped less NF (Figure 8b). This is at least partly due to the fact that GLC2000 mapped large areas in northwestern Canada as mosaic forests and Wang et al. [57] attributed this class to BF and grass PFTs (Table 4), although NF dominates most of those areas according to the other LC datasets (Figure 3).

The GLC2000 dataset mapped about 40% less fractions of crop than CCI, and it mapped more fractions of grass in the same area (Figure 8c,d). This is in contrast with the frequency comparison results showing that the percentages for crop from the GLC2000 and the CCI are within 10% (not shown). A comparison of the fractions for the main crop categories in the CW tables (Tables 4 and 5) for the GLC2000 and CCI datasets reveals the causes of this inconsistency. In our domain, the main crop class is LC16 from GLC2000 and it has a fraction of 0.5 for crop, while the rest is attributed to either grass or bare ground (Table 4). For the CCI dataset, the main crop class is LC11 and it is 100% attributed to crop (Table 5). Thus, smaller fractions of crop from GLC2000 than those from CCI are mainly due to the smaller fractions in their respective CW tables. In addition, GLC2000 mapped less fractions of grass in the arctic tundra than CCI.

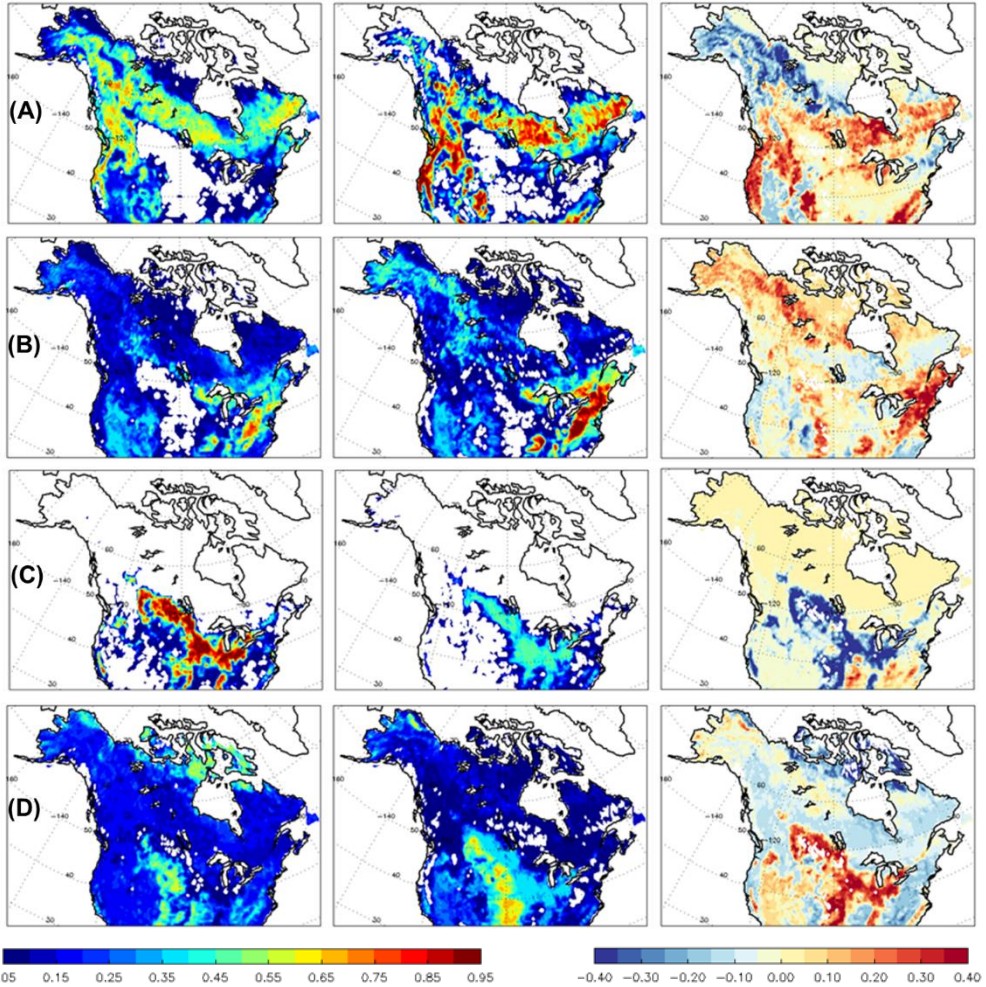

**Figure 8.** The fractional coverage of four CLASS PFTs from CCI (left), GLC2000 (middle), and the difference (GLC2000-CCI, right) for (**a**) needleleaf forest, (**b**) broadleaf forest, (**c**) crop, and (**d**) grass.

## 5. Discussion

The NALCMS LC dataset at 30 m resolution is used as the main reference data to assess the two global LC datasets in this study, which is not ideal as it is a satellite-derived product itself and subject to the limitations of such products (e.g., radiometric saturation, ambiguous spectral separability). We estimate the uncertainties in NALCMS by comparing it with the reference sample data, the EOSD, and the Hansen tree cover fraction datasets with regard to their capability in forest cover mapping. The results show that the wetland class in NALCMS exerts the largest uncertainty in forest cover mapping because treed wetland was not separated from herbaceous wetland in the legend. The inclusion of a taiga NF class with a lower tree cover threshold (>5%) than that used in the global datasets may also exert some uncertainty in NF mapping in NALCMS, which should be biased high relative to the other LC datasets in ecozones with a high percentage of taiga NF.

To compare the LC maps, we converted the maps into a 13-class common legend. The results show that using NALCMS as a reference, the 300 m CCI dataset provides much improved LC distribution over that from the 1 km GLC2000 dataset. Its more detailed LC classes and legend descriptions benefit the partitioning of LC classes into PFTs for use in LSMs. GLC2000 mapped too little NF in northwestern Canada, but too much in the Hudson Plains and some areas in northern Quebec, and it mapped a large area as MF and Mosaic categories, which results in large uncertainties for PFT mapping (i.e., unknown forest type). However, the CCI dataset appears to map more NF over Canada than all other LC datasets included in this study (e.g., Table 7). This is consistent with assessment results in previous

studies. Based on the GlobCover 2005 reference dataset, Tsendbazar et al. [20] showed that the CCI Phase I product (epochs 2005) had high producer (85.7%) but low user (58.9%) accuracies for evergreen needleleaf forest (ENF) (their Table 3). This over-mapping of ENF has not been improved much in the CCI Phase II product (used in this study) as shown in the user manual [12]. Based on the GlobCover 2009 reference dataset, the producer/user accuracies for ENF were found to be either 78%/64% for all sites or 85%/66% for homogeneous sites (Tables 3–6 and 3–7 in [12]). Investigation of the causes of this ENF over-mapping suggests the following. Firstly, the main classification process of the CCI LC chain relies on MERIS full resolution mean seasonal surface reflectance composites [59], whose number and period lengths vary regionally. In Canada and other high latitude regions (including most of the boreal forest regions where ENF dominates), usually only one 4-month seasonal composite centred on mid-July is available due to the long cloud and snow cover periods [48], which may have contributed to mislabeling other forest types to ENF and thus the over-mapping. Secondly, values in Tables 3–6 and 3–7 in [12] suggested that this was mainly due to confusion with mixed forests (90), shrubs (120) and grass (130). These LC classes all have low mapping accuracies in current global LC maps, likely due to the fact that most landscapes in these environments are a mixture of different types of vegetation [19,20]. We do not separate ENF from DNF (deciduous needleleaf forest) in this study for DNF accounts for less than 1% of NF over Canada according to the CCI dataset.

We have attempted to assess changes in the LC classes and forest cover by comparing the datasets between 2000 and 2010. The results show that relative to the MODIS maps, the CCI maps show smaller changes and/or changes in the opposite direction for most LC classes. The CCI maps indicate a forest gain while both the MODIS and Hansen datasets indicate a forest loss in most ecozones across Canada during the 2000 to 2010 period. This is consistent with results shown in [60]. Their explanation was that small-scale changes were not captured in the CCI product due to the use of coarse-resolution 1 km data for change detection before 2003 (300 m data were used after 2003), which was also acknowledged in the user manual [12].To verify this we include a comparison of changes in forest cover between 2004 and 2010 from the MODIS and CCI datasets (dashed lines in Figure 5). The results show close agreement for changes from CCI and MODIS in EZ1-6 and 8–10, while there are still large differences for the remaining ecozones. On one hand, this confirms that the use of coarse-resolution data for change detection before 2003 does contribute to large uncertainties in the CCI datasets. On the other hand, it suggests that changes in CCI are not always consistent with those from other datasets especially in northwestern Canada (EZ11, 14–18).

Changes in forest cover from the Hansen dataset are consistent with those from MODIS over the 2000 to 2010 period. However, Hansen shows more forest cover relative to both the reference samples and the NALCMS and EOSD datasets in EZ7 and 10, suggesting that the Hansen tree cover fraction dataset overestimates tree cover fraction, at least for tree cover fraction less than 25% (the threshold used for their forest definition), in wetland environments. Nevertheless, we demonstrate that the Hansen dataset is useful in assessing the performance of LC datasets in forest cover mapping. Given the limitations in currently available LC datasets and the importance of discriminating tree cover from short vegetation and bare ground in LSMs, an accurate vegetation continuous field tree cover fraction product would be very useful to inform the partitioning of LC classes into PFTs. Similar products can also be derived from the Laser Altimeters (by estimating tree cover fraction above a certain height) onboard the newly launched Ice, Cloud, and Land Elevation Satellite-2 and the Global Ecosystem Dynamics Investigation missions, which will be an important addition to the optical satellite-based products, e.g., [61].

A sub-pixel fractional error analysis was carried out for the 300 m CCI dataset using the 30 m NALCMS dataset as a reference, which revealed the detailed composition of the CCI LC classes. This will be especially useful for determining appropriate PFT fractions for the MF and Mosaic LC classes, and thus to develop a CW table representative of LC characteristics of Canada. Similar analyses can be done for other regions where fine-resolution LC datasets are available. Eventually a CW table specific to regional climate and landscapes can be created for the whole globe.

We amalgamated the 10 PFTs in the default CW table provided in the CCI LC product user manual into four broad PFTs in CLASS, and compared them with those based on the GLC2000 dataset currently used to provide initial surface conditions in the ECCC models. The results show that there are large differences in all four PFTs based on the two datasets, which should significantly impact model simulations [62]. Some of the differences are attributable to differences in the LC classes from the two datasets; however, fractions in the CW table appear to play a bigger role in most cases. This is consistent with findings in [31]. Due to the large ranges in some of the LC classes (i.e., >15% for forest classes), there are large uncertainties in the current available CW table for the CCI dataset [58].

## 6. Conclusions

Previous studies demonstrated the necessity for developing higher than 1 km resolution LC products in order to improve LC mapping in heterogeneous landscapes and the transition zones where LC changes primarily occur, and this was reinforced by findings in this study. The applicability of the CCI LC dataset for use in CLASS is being assessed through comparisons with the GLC2000 dataset currently in use in the model and with finer resolution (~30 m) datasets over Canada. The assessment results show that in comparison with the finer resolution maps over Canada, the 300 m resolution CCI dataset provides much improved LC distribution over that from the 1km resolution GLC2000 dataset. The GLC2000 mapped too little needleleaf forest in northwestern Canada but too much in the Hudson Plains, and it mapped a large area as mixed and mosaic forest categories, which resulted in large uncertainties for PFT mapping.

However, the CCI dataset appears to overestimate needleleaf forest cover especially in the taiga-tundra transition zone of northwestern Canada. This may have partly resulted from limited availability of clear sky MERIS images used to generate the CCI classification maps due to the long snow cover season in Canada, as well as the generally low mapping accuracies for mixed trees, shrubs, and grass in current LC datasets. Assessment of changes in LC mapping suggests that the CCI annual time series often show contradicting results relative to other datasets, and therefore caution should be exercised when using the annual CCI time series, especially prior to 2003, to represent changes in LC distribution. Nevertheless, the detailed LC classes and finer spatial resolution in the CCI dataset provide an improved reference map for use in CLASS and other LSMs.

Comparison of PFTs derived from the two global datasets suggests that uncertainties in the current CW tables are a major source of the often large differences in the PFT maps, and should be an area of focus in future work. Results in this study also suggest that the sub-pixel error analyses and continuous vegetation field tree cover fraction products are potentially useful for reducing uncertainties in the CW table and thus PFT mapping.

**Author Contributions:** L.W. conceived and designed this research, and wrote majority of the manuscript; P.B. created the CCI CW table, participated in discussions about the research and contributed to the writing; D.P. reinterpreted the reference sample data and contributed helpful advice for understanding the datasets and results, E.C. designed the reprojection/regridding method for the datasets and contributed to some data analyses; C.L. investigated the causes of needleleaf forest over-mapping in the CCI product and contributed helpful advice for understanding the dataset. All authors provided helpful comments and suggestions which improved the manuscript.

**Funding:** This research received no external funding.

**Acknowledgments:** The authors would like to thank Vivek Arora and Joe Melton (ECCC) for helpful comments on an earlier version of the manuscript, Elyn Humphreys (Carleton University) for providing some field data and for helpful discussions, and Peter Toose (ECCC) for helping to interpret images in Google Earth Engine. The authors wish to thank the anonymous reviewers for helpful comments and suggestions. Wulder's participation in this work was informed by "Earth Observation to Inform Canada's Climate Change Agenda" project jointly funded by the Canadian Space Agency, Government Related Initiatives Program, Climate Change Impact and Ecosystem Resilience, and the Canadian Forest Service of Natural Resources Canada.

**Conflicts of Interest:** The authors declare no conflict of interest.

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
