# Peer review of "Comparison and Assessment of Regional and Global Land Cover Datasets for Use in CLASS over Canada"

_remotesensing, doi:10.3390/rs11192286_

Round 1

Reviewer 1 Report

Line 308  - I think this might be a mistake – would you combine those EOSD classes into the forested class – or open and sparse into NF and the dense NF into forested?     "The 23 classes in EOSD can be easily merged (e.g. combine classes of conifer-dense, conifer-open, and conifer-sparse into NF) and linked to the broad categories in the common legend (Table 3)."

Line 348  awkward wording – could use semi-colon and link to previous sentence. "On account of this, we compute the producer/user accuracies for 10km by 10km pixel blocks."

Line 354 – personal preference – the reader should not have to go to references. "Sub-pixel fractional error matrices were introduced by [16]."  

Line 372.  The sentence says there are 9 vegetation PFTs.  In Table 4 – there are 8 columns in addition to the GLC2000 legend description.  Of those four are vegetated classes and four non-vegetated types.  Should the sentence say 8 PFTs, should there be another column or should this be described differently?  Based on further reading it seems as though the 9 vegetation PFTs are not included in the table but are indirectly referenced here.  It would be good to make that clear.  "Based on expert knowledge of global biomes and class descriptions, Wang et al [57] created a cross-walking table to convert the GLC2000 22 LC classes to nine vegetation PFTs for use in CTEM, which were combined onto the four broad PFTs recognized by CLASS (Table 4)."

Table 6.  Should be “number of forest samples” and "number of non-forest samples".

Figure 2. Should be a bar chart or scatter plot or table – the current graph implies a trend between the ecoregions.

Line 452 – Personal preference – the reader should not have to refer to citations.  An alternative would be to remove “after” and keep the citation. "EOSD mapped at least 100% more wetland than any other LC product, likely due to the different definition of wetland in its legend (as wetland-treed is a treed/forest category, after [52])."

Figure 5. Should be a bar chart or scatter plot – or the curves should reflect the change through time with ecoregions being independent.

Figure 8 should include the legend for the color ramp or refer to the previous figure with that ramp.

Author Response

Thank you for reviewing the manuscript.

Line 308 - I think this might be a mistake – would you combine those EOSD classes into the forested class – or open and sparse into NF and the dense NF into forested?     "The 23 classes in EOSD can be easily merged (e.g. combine classes of conifer-dense, conifer-open, and conifer-sparse into NF) and linked to the broad categories in the common legend (Table 3)."

We meant to combine the corresponding classes, for example, combine classes of conifer-dense, conifer-open, and conifer-sparse into NF. As an additional clarification, NF is needleleaf forest (not non-forest) in case there was confusion. To clarify, we have changed the sentence to the following:

"The 23 classes in EOSD can be easily merged to the corresponding broad categories in the common legend (e.g. combine classes of conifer-dense, conifer-open, and conifer-sparse into needleleaf forest (NF))"

Line 348 awkward wording – could use semi-colon and link to previous sentence. "On account of this, we compute the producer/user accuracies for 10km by 10km pixel blocks."

We changed it to the following:

“Therefore, we compute the producer/user accuracies for 10 km by 10 km pixel blocks.”

Line 354 – personal preference – the reader should not have to go to references. "Sub-pixel fractional error matrices were introduced by [16]."

We modified the sentence:

"Sub-pixel fractional error matrices were introduced by Latifovic and Olthof [16]."

Line 372. The sentence says there are 9 vegetation PFTs. In Table 4 – there are 8 columns in addition to the GLC2000 legend description. Of those four are vegetated classes and four non-vegetated types. Should the sentence say 8 PFTs, should there be another column or should this be described differently? Based on further reading it seems as though the 9 vegetation PFTs are not included in the table but are indirectly referenced here. It would be good to make that clear. "Based on expert knowledge of global biomes and class descriptions, Wang et al [57] created a cross-walking table to convert the GLC2000 22 LC classes to nine vegetation PFTs for use in CTEM, which were combined onto the four broad PFTs recognized by CLASS (Table 4)."

Note we only included the four CLASS PFTs in Table 4. We mention the nine PFTs in CTEM because they were created by Wang et al [57], and were used to produce the four CLASS PFTs.

Table 6. Should be “number of forest samples” and "number of non-forest samples".

Done.

Figure 2. Should be a bar chart or scatter plot or table – the current graph implies a trend between the ecoregions.

We have changed it to a bar plot.

Line 452 – Personal preference – the reader should not have to refer to citations. An alternative would be to remove “after” and keep the citation. "EOSD mapped at least 100% more wetland than any other LC product, likely due to the different definition of wetland in its legend (as wetland-treed is a treed/forest category, after [52])."

Done.

Figure 5. Should be a bar chart or scatter plot – or the curves should reflect the change through time with ecoregions being independent.

We have changed it to a bar plot.

Figure 8 should include the legend for the color ramp or refer to the previous figure with that ramp.

We have added the color ramp.

Reviewer 2 Report

The objective of this study is to assess and compare the CCI and the GLC2000 datasets with high-resolution (~30m) land cover products (NALCMS, EOSD LC) over Canada, to better understand their patterns of agreement and disagreement, and to determine the applicability of the CCI dataset for use in CLASS. The work was exhaustive and the paper as a good discussion and is clear. However, this paper does not fit sufficiently in the scope of remote sensing, nor present a significant novelty. Hence, I recommend their publication in a different journal.

Author Response

Thank you for reviewing the manuscript.

The objective of this study is to assess and compare the CCI and the GLC2000 datasets with high-resolution (~30m) land cover products (NALCMS, EOSD LC) over Canada, to better understand their patterns of agreement and disagreement, and to determine the applicability of the CCI dataset for use in CLASS. The work was exhaustive and the paper as a good discussion and is clear. However, this paper does not fit sufficiently in the scope of remote sensing, nor present a significant novelty. Hence, I recommend their publication in a different journal.

Although the primary objective of the paper is to assess the applicability of CCI dataset for use in CLASS, the main results are the comparison and assessment of the satellite-derived land cover datasets. There are no model simulation results in the paper. We believe the paper fits the scope of Remote Sensing well.

Reviewer 3 Report

The importance of Land Cover Information is undisputed and the need for higher-resolution land cover products (< 1 km) have been demonstrated in recent work, which could be confirmed by this study by comparing a 300 m CCI dataset with the 1000 m GLCGLCGGGGGLC2000.

The study is well structured and organized with a clear presentation of results. It is well-written and provides a sound methodological approach which is in particular of importance for this work as the investigation area is limited to Canada and focused on the national CLASS. Due to the methodological approach, however, the study will reach a larger readership.

The comments I have are therefore limited and I would recommend publication after minor modifications concerned with the presentation style:

I feel that the second part of the abstract could be condensed as it presents a number of results that feel a bit too narrow for an abstract that attempts to address (and eventually attract) a broader readership. Given that the comparably brief conclusions are not much different from the abstract, one could try to communicate the findings in a more general way perhaps. But I will leave this to the authors to decide.

L 35: you might want to explain what a "good" reference map is?

L 61: what are "improved computing opportunities"?

Fig 1: ecozone label are illegible, legend is too small, the figure could be presented at larger scale.

Fig 1 caption: NALCMS is explained in the text later (L. 139), it would be helpful to also explain in the caption

L 129: EZs have not been explained/listed anywhere in a comprehensive way. Here and there, the text jumps between ID and labels. It would be helpful to have a table to which the text refers.

L 176: I haven't checked but the processing approach that you mention is provided in [43]? If not, more details on "hyperclustering, cluster merging and labelling" would be appreciated as it does not say much.

Tbl. 1: check use of parentheses vs. brackets.

Tbl 2: see comment above. Much needed and a more comprehensive/clearer labelling of column 1 would be appreciated.

L 339: citation style (Manandhar et al., 2009) -> [53]

L 417: what about EZ3? Wheredo we find what EZ16 is (see comment above)

L 419: can you shed some (quantitative) light on why 10% are acceptable? Would 20% also be acceptable given the different methods used?

L 432 (caption Fig 2): obtain(-ed) from...

Fig 3: illegible due to size. Perhaps also move legend text to legend bar?

Fig 6 caption: could you add a percentage sign to the legend to reflect accuracies

L 533: the style of presentation of results (NF 0.23 etc.) is not traceable for the reader. Where are you exactly referring to?

Tbl 8: Could classes be described more comprehensively? It is really a bit hard to find some of the ecozones and landcover information spread across the manuscript -- in particular as you are switching back and forth between IDs and labels. A more consistent display throughout the manuscript would be extremely helpful.

Please make sure to introduce a space between numerical values and units (L. 51, 55, 66, 71, 103, 107, 138, 149, 153, and so on), some for Fig.X labels and references.

Author Response

The importance of Land Cover Information is undisputed and the need for higher-resolution land cover products (< 1 km) have been demonstrated in recent work, which could be confirmed by this study by comparing a 300 m CCI dataset with the 1000 m GLCGLCGGGGGLC2000.

The study is well structured and organized with a clear presentation of results. It is well-written and provides a sound methodological approach which is in particular of importance for this work as the investigation area is limited to Canada and focused on the national CLASS. Due to the methodological approach, however, the study will reach a larger readership.

Thank you for reviewing the manuscript.

The comments I have are therefore limited and I would recommend publication after minor modifications concerned with the presentation style:

I feel that the second part of the abstract could be condensed as it presents a number of results that feel a bit too narrow for an abstract that attempts to address (and eventually attract) a broader readership. Given that the comparably brief conclusions are not much different from the abstract, one could try to communicate the findings in a more general way perhaps. But I will leave this to the authors to decide.

We have made some changes to the second part of the abstract to make it more general as in the following:

"Global land cover information is required to initialize land surface and Earth system models. In recent years new land cover (LC) datasets at finer spatial resolutions have become available while those currently implemented in most models are outdated. This study assesses the applicability of the Climate Change Initiative (CCI) LC product for use in the Canadian Land Surface Scheme (CLASS) through comparison with finer resolution datasets over Canada, assisted with reference sample data and a vegetation continuous field tree cover fraction dataset. The results show that in comparison with the finer resolution maps over Canada, the 300 m CCI product provides much improved LC distribution over that from the 1 km GLC2000 dataset currently used to provide initial surface conditions in CLASS. However, the CCI dataset appears to overestimate needleleaf forest cover especially in the taiga-tundra transition zone of northwestern Canada. This may have partly resulted from limited availability of clear sky MEdium Resolution Imaging Spectrometer (MERIS) images used to generate the CCI classification maps due to the long snow cover season in Canada. In addition, changes based on the CCI time series are not always consistent with that from the MODIS or a Landsat-based forest cover change dataset, especially prior to 2003 when only coarse spatial resolution satellite data were available for change detection in the CCI product. It will be helpful for application in global simulations to determine whether these results also apply to other regions with similar landscapes, such as Eurasia. Nevertheless the detailed LC classes and finer spatial resolution in the CCI dataset provide an improved reference map for use in land surface models in Canada. The results also suggest that uncertainties in the current cross-walking tables are a major source of the often large differences in the plant functional type maps, and should be an area of focus in future work."

L 35: you might want to explain what a "good" reference map is?

We have changed “good” to “improved”, which is more appropriate here.

L 61: what are "improved computing opportunities"?

We have changed it to “advanced computing power”.

Fig 1: ecozone label are illegible, legend is too small, the figure could be presented at larger scale.

We have reproduced the figure at a larger scale and using larger font and more contrast colors for the ecozone labels.

Fig 1 caption: NALCMS is explained in the text later (L. 139), it would be helpful to also explain in the caption

Done.

L 129: EZs have not been explained/listed anywhere in a comprehensive way. Here and there, the text jumps between ID and labels. It would be helpful to have a table to which the text refers.

We have modified Table 2 to provide a unique name for each ecozone used in the paper.

L 176: I haven't checked but the processing approach that you mention is provided in [43]? If not, more details on "hyperclustering, cluster merging and labelling" would be appreciated as it does not say much.

Yes, [43] provides details of the processing approach.

Tbl. 1: check use of parentheses vs. brackets.

The parentheses in the CCI legend are used to indicate sub-classes, while the brackets show the main classes. We have added this explanation in the caption.

Tbl 2: see comment above. Much needed and a more comprehensive/clearer labelling of column 1 would be appreciated.

We have extended text in column 1 to provide a clearer labelling of the EZs as suggested.

L 339: citation style (Manandhar et al., 2009) -> [53]

Done.

L 417: what about EZ3? Wheredo we find what EZ16 is (see comment above)

We have increased the scale and the font size for labels of Figure 1. Forest cover agrees well between Hansen and NALCMS in EZ3, but it is lower in EOSD. This was summarized in the last sentence of the paragraph: “The proportion of forest cover from EOSD tends to be lower than that from Hansen and NALCMS in most ecozones across Canada.”

L 419: can you shed some (quantitative) light on why 10% are acceptable? Would 20% also be acceptable given the different methods used?

This is subjective. We have changed “acceptable” to “understandable”.

L 432 (caption Fig 2): obtain(-ed) from...

Done.

Fig 3: illegible due to size. Perhaps also move legend text to legend bar?

We have increased the fonts for all the captions.

Fig 6 caption: could you add a percentage sign to the legend to reflect accuracies

Done.

L 533: the style of presentation of results (NF 0.23 etc.) is not traceable for the reader. Where are you exactly referring to?

We have included the names of the 13 LC classes in the caption of Table 8.

Tbl 8: Could classes be described more comprehensively? It is really a bit hard to find some of the ecozones and landcover information spread across the manuscript -- in particular as you are switching back and forth between IDs and labels. A more consistent display throughout the manuscript would be extremely helpful.

We have added the names of each of the 13-class in the caption of Table 8.

Please make sure to introduce a space between numerical values and units (L. 51, 55, 66, 71, 103, 107, 138, 149, 153, and so on), some for Fig.X labels and references.

Done.